# "Pray Aggressively for a Higher Goal—The Unification of All Christianity": U.S. Catholic Charismatics and Their Ecumenical Relationships in the Late 1960s and 1970s

## Valentina Ciciliot

Department of Humanities, Ca' Foscari University of Venice, 30123 Venezia, Italy; vciciliot@unive.it

**Abstract:** In July 1977, 50,000 Christians from different backgrounds and traditions converged on Kansas City to participate in the Conference on Charismatic Renewal in the Christian Churches. Catholic charismatics played a key role in its organization, relying on all their ecumenical contacts built since the origins of the Catholic Charismatic Renewal (CCR) in 1967 at Duquesne University in Pittsburgh (PA). If the Kansas City conference represented the zenith of a shared unified vision for all charismatic Christianity, it also showed the emergence of the crisis which affected Catholic charismatic communities and their connection with Rome. This paper will explore U.S. Catholic charismatics' relationships with other Christian denominations and groups in the initial development of the CCR, particularly in structuring Catholic charismatic communities, and their ecumenical perspectives in the tension between needs for legitimization (by the Vatican) and needs for self-expression.

**Keywords:** American Catholicism; Catholic Charismatic Renewal (CCR); Charismatic movement; ecumenism; Kansas City conference; Shepherding movement





## 1. Introduction

The Catholic charismatic renewal (CCR) officially began in February 1967, at Pittsburgh's Duquesne University (PA), when a history professor, William (Bill) Storey, and a graduate student, Ralph Kiefer, claimed to be baptized in the Holy Spirit in a charismatic prayer group of Episcopalians. Through personal contacts, this experience of the Holy Spirit soon spread to the University of Notre Dame (Notre Dame, Indiana), then to Michigan State University (East Lansing, Michigan), to the University of Michigan (Ann Arbor, Michigan), and many other parts of the United States.[1] At ever-increasing numbers of locations, regular prayer meetings, and sometimes covenant communities, developed.[2] In less than a decade, the CCR went beyond U.S. national borders, becoming a worldwide movement within the whole Catholic Church with active functioning structures and supportive officials in Rome.

In the initial development of the movement, the University of Notre Dame in Indiana played a key role, not only in terms of organizational work—as an example, the Catholic Charismatic Renewal Service Committee (CCRSC, later renamed National Service Committee, NSC) was established there[3]—but also of visibility. In fact, the considerable and rapid shift of the charismatic activities at this university to the public arena was the result of the national publicity received thanks to the attention of the media, which also aroused the interest of the U.S. charismatic Protestant world.[4] In 1967, during the initial months of charismatic "experiments" at the university, Doug Wead, at that time a Pentecostal leader, was sent to one of the prayer meetings on campus by his father, the pastor of Calvary Temple in South Bend, to understand what he perceived as the "paradox of 'Catholic Pentecostals'".[5] In appreciation and incredulity, he reported:

> There was an excitement in my first charismatic Catholic experience that I have never recaptured. For me, there was at least one unique factor. Somehow, God had changed. Suddenly. He was more than a conservative Republican from northern Indiana. He became a God of many people, people of different cultural,

ideological and racial backgrounds. This university community in which I first saw the Catholic renewal was urban and liberal—the opposite pole ideologically. Outside of football enthusiasm for the Fighting Irish we had nothing in common. Yet, while I was busy across town working at Calvary Temple (which I assumed to be God's South Bend headquarters), God had been very busy at Notre Dame.[6]

There was a historic animosity between Pentecostals and Catholics, but this, nevertheless, left room for rapprochement and further dialogues that would be constitutive of the origins and early developments of the Catholic charismatic movement in the United States. As a matter of fact, for the first time in the history of the Catholic Church, a form of spirituality coming from Protestantism, and to all appearances quite distant from the typical Catholic identity, was experienced and reworked by Catholics.[7]

The purpose of this article is precisely to highlight the experiences that came with the contacts between charismatic Catholics and Protestants at the beginning of the CCR. As the "insiders" within the Catholic charismatic renewal sought to legitimize the movement, their historical accounts, and even more, the first theological declarations, preferred to focus mainly on the "overlooked" charismatic aspects of the Catholic tradition, in order to portray the renewal as something "internal" or "constitutional" to the church, thus anchoring it to Catholic ecclesiology and biblical exegesis.[8] It suffices to think about the constant references to Pope Leo XIII and his encyclical *Divinum illud munus* (1897); the blessed Elena Guerra (1835–1914), who has been considered a precursor of the charismatic movement;[9] Pope John XXIII and his prayer for a "new Pentecost"; and the Second Vatican Council.[10] As follows, the abundant interconnections between Catholic charismatic and Protestant leaders were put in the background, if not partially censored, for the sake of acknowledgement and acceptance.

In the 1970s and 1980s, these unheralded relationships among U.S. charismatic leaders across denominational and non-denominational institutions were influential and fruitful and able to shape not only the U.S. CCR but also its subsequent worldwide development. As an example, some of the ecumenical activities promoted by U.S. Catholic charismatic communities in those decades, which were ultimately the result of that intra-Christian charismatic dialogue in the U.S., did not find favor in Rome and eventually became one of the targets of the "Catholicization" process of the charismatic renewal during John Paul II's pontificate.[11]

Ever since the beginning of the 1960s, mainline Protestant churches in the U.S. were experiencing the charismatic "wave", which would soon interact with the Catholic world.[12] In fact, in Pittsburgh Catholics, who eventually organized the first official charismatic encounter at Duquesne University, received the baptism in the Spirit thanks to a small charismatic Episcopalian group.[13] However, even though some of the classical Pentecostals at that time were no longer particularly charismatic, those who still were taught Catholics a lot. Circulation of texts such as charismatic Pentecostal David Wilkerson's *The Cross and the Switchblade* (written with John and Elizabeth Sherrill) and John Sherrill's *They Speak in Other Tongues* was (I believe it is "were" because subjects are Wilkerson's and Sherill's books . . . ) considered inspiring and life-changing by the first Catholic charismatic leaders.[14] In addition to denominational contacts, non-denominational Pentecostal and charismatic organizations such as Demos Shakarian's Full Gospel Businessmen's Fellowship International (FGBMFI) provided an important platform for Catholic charismatic renewal experiences. Founded in 1951, the FGBMFI was a fellowship of business people with the goal to merge faith, particularly charismatic, with business practices. Its periodical "Voice" carried regular reports of pastors and laity who claimed to have experienced baptism in the Holy Spirit, while conferences organized in its many regional chapters throughout the United States contributed toward spreading the practice.[15] For example, the earliest Notre Dame and South Bend Catholic charismatics went to the house of a Calvary Temple deacon who was also president of the local South Bend chapter of the FGBMFI to learn about the charismatic gifts of the Spirit.[16] This is to say that inter-connections across denominations on charismatic spirituality were as intellectual as practical, and they deeply influenced part

of the U.S. Catholic charismatic groups, particularly those related to the Midwest area and interested in building charismatic communities.

Thus, this article intends to focus on three aspects that have been less explored by historiography to demonstrate how charismatic ecumenical contacts and networks between the late 1960s and 1970s unquestionably shaped the Catholic charismatic renewal in the United States and worldwide:[17] the role of Episcopalian pastor Graham Pulkingham and his Church of the Redeemer in Houston (Texas) in modeling the first Catholic charismatic covenant communities in the late 1960s; the relationships between Catholic charismatic and non-denominational charismatic leaders, particularly those involved in the Shepherding movement, in the 1970s, for the sake of a pan-charismatic project; and finally, the 1977 Kansas City Conference as the zenith but also the beginning of the crisis of the pan-ecumenical/interdenominational charismatic experience.

## 2. Graham Pulkingham and the Church of the Redeemer

From its beginning, the CCR expressed two forms of practicing charismatic spirituality—prayer groups and communities—differing mainly on the degree of active involvement. Looking for a way to live out a more charismatic Christian life and to provide services to the fledgling movement between 1970 and 1971, three communities roughly in the same geographical area (Indiana and Michigan) were formed: The Word of God (TWOG) in Ann Arbor, followed by True House (TH) at Notre Dame, and the People of Praise (POP) in South Bend. Although their stories turned out differently,[18] they were initially inspired by early Christian communities and originally shaped by similar visions, particularly TWOG and POP, which were both ecumenical. As covenant communities, which were at that time something ecclesiastically new within the Catholic Church, members bound themselves to one another by a solemn agreement called a "covenant", thus maintaining affiliation in their own church or denomination while also belonging to the covenant community. Soon, related communities grew up in Newark, Augusta, Minneapolis, Pittsburgh, Phoenix, and several other places, whereas in Dallas and Cincinnati, communities not related to this tradition were founded as well.

If such communities were influenced by some of the communitarian ideas expressed by the counterculture of the 1960s, they took their inspiration and many of their initial ideas and practices from the Episcopal Church of the Redeemer in Houston, Texas, led by Graham Pulkingham. Pulkingham was a charismatic Episcopal pastor who, thanks to his encounter with David Wilkerson, revolutionized his life and founded a charismatic community within the parish of the Church of the Redeemer in 1964. Out of this community grew an international ministry of preaching, community living and music, thanks chiefly to Pulkingham's healing ministry and attractive personality, and this parish church soon became a charismatic pilgrimage site for the entire charismatic world.[19] As an example of the impact of the Church of the Redeemer on other Christian denominations and generally in people's lives, Anglican priest Michael Harper, who eventually became one of the leaders of the charismatic renewal in Britain and founder of the ecumenical agency Fountain Trust, arrived in Houston in February 1966, and was so influenced by his experience there that it changed his ministry forever.[20]

Although the first Catholic charismatic literature often omitted to quote Graham's influence in the community-building process, his relations with Catholic charismatic leaders of the first covenant communities are undeniable. They were prone to involve Pulkingham from the beginning, inviting him to speak at the 1969 Notre Dame conference and to collaborate with Catholic charismatic "New Covenant" magazine as well. (here I add a footnote–I am not able to put it in the correct way: See as examples: "Marriage, Community, Service. Interview with Rev. Graham Pulkingham." *New Covenant*. December 1971, pp. 11–14; Graham Pulkingham. "Headship in Christian Marriage." *New Covenant*. December 1972, p. 11; Id. "The Fisherman, INC" *New Covenant*. March 1973, pp. 5–6. It seems that Pulkingham "endorsed the fledging Catholic charismatic magazine *New Covenant* as a publication that agreed with the aims of Redeemer and the Fishermen [a.n.

a charismatic consultant agency], to the point the Redeemer even decided not to begin its own publication because *New Covenant* was already doing the job".[21] In 1970, two members of Redeemer went to Ann Arbor for a month, staying with Ralph Martin, Steve Clark, Jim Cavnar, and Gerry Rauch, the four TWOG founders. The relationship between the two communities (that in Ann Arbor, in fieri) was mutual: "Word of God was good at organizing and getting things done, and they would end up being major systematizers of community and charismatic renewal. But the Redeemer people know how to love and lay down their lives for each other, how to make community a life-transforming event, which impressed the Michigan visitors no end. Word of God people got the idea of extended-family households, which were mainly singles living with Redeemer families, from Redeemer".[22]

Remarkably, in connection with this interdenominational rapprochement, he visited South Bend before the foundation of People of Praise and met informally with several leaders, in a key moment for the developing community. The idea of establishing a formal community was already under investigation, but Pulkingham imbued the leaders with more confidence and was helpful in showing them a somewhat different perspective, where, among other things, prophecy was more central. Although oral memories are not unanimous in defining the exact time, Pulkingham probably visited South Bend between summer 1970 and summer 1971. Tom Noe, one of the leaders of the Christ the King prayer meeting in South Bend, involved in the True House community, and later a member of the POP, had already gone to the Church of the Redeemer and spoke with its pastoral leaders in summer 1969. The report of his trip provides a vivid picture of those days and shows the perception of a Catholic who looked with some admiration at the charismatic experience at the Redeemer:

> Tom Noe spent two weeks hitchhiking down through Texas and Ky. One of the women who picked him up is an ex-nun who has received the gift of prophecy, predication, healing and discernment, but not through any prayer group. She received them individually and Tom was able to talk with her about prayer groups and put her in contact with a group in Austin. The Lord seems to have supported Tom well in his travels. He reports spending only $1.50 the whole two weeks. While in Texas he was able to stop at Redeemer Parish, Rev. Graham Pulkingham's Episcopal Parish in Houston. [ . . . ] The Parish is completely Spirit-filled and the members constantly minister to one another. They get up in the morning and hug one another, saying "Praise the Lord!". They even hug one another throughout the day. A few weeks ago one member of the parish was arrested for driving with his hands in the air![23]

At the beginning of August 1971, Kevin Ranaghan and other local charismatic leaders acted on the formation of the community. A few months later, on 15 October 1971, a group of people established a formal covenant community, renamed on 27 February 1972, People of Praise.[24] Except for differences in timing, parts of the Church of the Redeemer community's features were imported by TWOG, TH, and POP, such as the household structures, St. Paul's teaching on "foundational gifts" and ministries, and the focus on prophecy,[25] in this case, showing in-depth spiritual and ecclesiological convergences between charismatic Episcopalians and charismatic Catholics.

### 3. Non-Denominational Charismatics: The Shepherding Movement

The Shepherding movement, or Discipleship movement, took its origins in the early 1960s in Fort Lauderdale, Florida, at the Shepherd's Church, when a specialized group of pastors—Bob Mumford, Charles Simpson, Derek Prince, Don Basham, and later Ern Baxter[26]—created the Holy Spirit Teaching Mission (HSTM), later Christian Growth Ministries (CGM), a missionary agency that first sought to establish a certain kind of charismatic leadership within Christian churches.[27] However, it emerged as a distinct non-denominational movement in 1974. They advocated the need for what they described as "discipleship", thorough personal pastoral care or shepherding care, thus theorizing that every believer

needed to submit to a spiritual authority who would help the individual develop Christian maturity within a covenant relationship. Mutually, all pastors and leaders needed to be personally submitted to another leader to foster accountability, in "a kind of a chain of command, with a senior or presiding pastor overseeing the local church network of pastoral leaders".[28] Each presiding pastor was ultimately submitted to one of the four—later five—teachers of Fort Lauderdale. This pyramidal system was soon able to create translocal pastoral relationships. If, at the beginning, the aim was to spread the movement within existing churches, after the charismatic renewal took hold in different denominations, the aim changed to creating independent charismatic church structures under the Fort Lauderdale leadership, if not control. In fact, the Shepherding movement was non-denominationally pan-charismatic, well-equipped, well-organized, and media-oriented. Its videotapes and popular magazine "New Wine", with a monthly circulation of 110,000 copies in 1976,[29] spread the discipleship concept all over the charismatic world.

In 1975, response to the growing influence of this movement precipitated a controversy within the charismatic movement as a whole, as several charismatic leaders expressed public disagreement with the Fort Lauderdale group, mainly fearing the establishment of a new large charismatic independent church. According to Harper, this dispute was "far and away the most disturbing controversy to hit the Charismatic Renewal".[30]

The CGM annual outreach, the National Men's Shepherds Conferences, fueled the controversy, which became a very public debate in 1975 and 1976, thanks to the media attention on it and the involvement of prominent Christian leaders such as Episcopalian Dennis Bennett, Lutheran Harald Bredesen, charismatic publisher Dan Malachuk, and evangelical Pat Robertson.[31] At stake there were not only the competing meanings of "authority" within the different Christian traditions, but also the role of charismatics within their own churches and within the broader scenario of U.S. Christianity. To handle this intense conflict, a special ad hoc charismatic leaders' committee was established. Its first "resolution" meeting was held in Minneapolis on 9 August 1975, where the opposition to the CGM was mainly from independent charismatics, particularly Pentecostals, in primis the FGBMFI leaders, fighting a sort of "turf war". In fact, along with the concerns of doctrinal and practical extremes, the majority of the Shepherding movement's adherents came from independent Pentecostal/charismatic sectors.[32] Denominational charismatics appeared to be less disturbed with the CGM teaching. For example, the controversial discussion on the spiritual authority claimed by the Fort Lauderdale leaders versus ecclesiastical authority— could a person be submitted to his own local church leadership and also to a shepherd outside that church?—had to a certain extent already been addressed within Catholic charismatics by reference to the traditional Catholic concept of spiritual directorship, which had been used as a model within covenant communities.[33] Moreover, "perhaps the security and the heritage of their denominational affiliations made the Shepherding movement less threatening to these mainline Charismatics", being less concerned about the movement to become a new denomination.[34]

For the purposes of this article, what is interesting is not only that prominent Catholic charismatic community leaders such as Ralph Martin, Steve Clark and Kevin Ranaghan, and also ecumenicist and theologian Killian McDonnell, were involved even before the beginning of the controversy, but also that the second resolution meeting was held in Ann Arbor on 16–17 September 1975,[35] proving how much the place was important in the charismatic ecumenical world, how respected Catholic charismatics were as mediators in the controversy, and how deep the ongoing relationship between Ann Arbor-South Bend and the Fort Lauderdale leaders was. The relationship between Catholics Clark, Martin, Paul DeCelles (POP), Ranaghan, and Lutheran Larry Christenson, and the CGM leaders is also proven by regular meetings—two or three times a year—held beginning in 1974 with the aim of uniting the charismatic communities. The relationship of the "Ft. Lauderdale brethren" and the leaders of the Catholic charismatic communities, however, predated 1974, when by a crisis in the HSTM, the group begun working on national "men's shepherds" conferences, designed to teach the functioning of pastoral authority and to

form charismatic leaders in the United States and around the world. These leaders, in fact, organized the 1974, 1975 and 1976 National Men's Shepherds Conference, respectively, in Montreat, Kansas City and Atlanta.[36] This group of people referred to themselves as the "Council", the "General Council", and finally, as the "Ecumenical Council".[37] The controversy mentioned earlier occurred in the midst of these project meetings, and it was in the background for most of the life of the Ecumenical Council. Although more research should be conducted on the history of this body, from the minutes of the Ecumenical Council, it clearly emerges that these Christian leaders had established a mutual dialogue and cooperation, so much so that at a certain point, they thought of merging their communities into a single one,[38] being acknowledged beyond the circle of its leaders, despite the secrecy of part of these meetings.[39] Strong elements of discipleship and pan-charismatism were present within the Council. For example, leaders submitted to one another[40] and while maintaining their various denominational differences, emphasized charismatic spirituality almost as an exclusive trait for Christian unity. Through Martin's and Clark's intercession, the Ecumenical Council also had contact with Cardinal Suenens, who was, at that time, the advisor of the CCR for the Catholic Church: the entire group travelled to Belgium and the Holy Land in 1977 to discuss ecumenism and their relationship with the cardinal and his charismatic circles. Particularly significant was the idea of establishing a sort of training center for charismatic leaders that seemed to be supported by Suenens:

III.   Joint Project Discussion: A. Possible Leadership Training Institute:

1. Three-month training for young men to be sent into the ministry.
2. Cardinal Suenens is interested in a training institute where men could be ordained into the R.C. priesthood and diakonate [sic!].
3. Training to consist of three 3-month sessions for three years. A. Some study at home between 3-month terms. B. Training program to be administered by community heads. C. Teaching by recognized teachers. D. Could be held in different locations each year [ … ]
6. Training Institute would train not only elders and shepherds, but also teachers.
7. What is the real objective (philosophy) for such an Institute? A. To affect the spiritual and academic leadership of the Church. B. To bring unity and compatibility between emerging communities. C. To train and equip teachers for the Body of Christ. D. Such training would provide protection for the Body of Christ. E. To help deprogram ministers with only an academic mind-bent and increase their ability to discern spiritually when confronted by worldly expertise and wisdom. F. To increase the ability of leaders to communicate effectively. [ … ]
12. Whose Institute is it? A. It would be founded and directed by the General Counsel. B. Derek Prince, Steve Clark and Larry Christenson appointed as committee to brainstorm the concept and come up with specific proposals for forming the Training Institute. We work with Cardinal Suenens, but our communities would be in charge. Committee to report at December Council meeting."[41]

Although the idea of the training center was soon abandoned, supposedly because of Catholic internal tensions between Martin, Clark, and Suenens,[42] this project shows a deep cross-sectional interconnection between high-profile charismatic leaders regarding important ecumenical aspects such as leadership training and the pan-charismatic network. Eventually, tensions in the late 1980s between Ann Arbor and South Bend leaders resulted in Ranaghan's and DeCelles' resignation from the Council[43] and in February 1985, it was dissolved by mutual consent.[44]

While the discipleship controversy, during which Catholic leaders mostly advocated for the five teachers, formally ended in March 1976, at the meeting in Oklahoma City, through the establishment of the Charismatic Concerned Committee (CCC), with the aim to resolve possible future charismatics' controversies, the Shepherding movement continued to affect charismatic churches throughout the 1970s and 1980s, to a certain extent affecting U.S. Catholic covenant communities as well, to the point that authority management

and submission relationships were the main topics scrutinized by Catholic ecclesiastical authorities in the 1990s during several official investigations of such communities.[45]

## 4. The Kansas City Conference

It was the relationship between Catholic charismatic and CGM leaders that gave rise to the largest ecumenical event ever seen in the United States, the Kansas City Conference on the Charismatic Renewal in the Christian Churches (CCRCC), in Missouri, on 20–24 July 1977.[46]

Already in the 1974 Montreat Men's Shepherds Conference Ralph Martin delivered a key message entitled *The Mighty Stream of God*, where he conceptualized the idea of three rivers of charismatic renewal: the Classical Pentecostals, the Protestant Charismatics, and the Catholic Charismatics, and the need for them to "flow together".[47] This theorization was the starting point for the Kansas City conference, which sometimes was also called the three-rivers conference.[48] The official planning committee, whose chairman was Kevin Ranaghan, was composed of fourteen charismatic leaders among the main Christian churches affected by charismatic spirituality: Lutheran Larry Christenson, Presbyterian Brick Bradford, Episcopal Robert Hawn, Baptist Roy Lamberth, Baptist Ken Pagard, Mennonite Nelson Litwiller, Methodist Ross Whetstone, Elim Pentecostal Carlton Spencer, Foursquare Pentecostal Howard Courtney, International Pentecostal Holiness Vinson Synan, Church of God in Christ Pentecostal Ithiel Clemmons, Messianic Jewish David Stern, independent charismatic Bob Mumford and Robert Frost (later replaced by Judson Cornwell).[49] The CCRSC corporation—Charismatic Renewal Services (CRS)—was chosen to administer and to own the conference, mainly for its financial availability at that time, even though funds from other charismatic institutions were raised. Soon, an executive committee to advise the chairman, following Martin's idea of the "three streams" concept, was added, with Christenson for the mainline Protestants, Synan for Classical Pentecostals and Ranaghan for Catholics as members.

The core theme of the conference was "Jesus is Lord", representing both the essential charismatic Christian spiritual motif and the aim of bringing together all the streams of Pentecostal and charismatic renewal in the United States, or, as Synan detailed, "from as disparate as fire-breathing Southern fundamentalist Pentecostals to sophisticated European Roman Catholic prelates".[50] Thus, both ecumenical "cross-pollination"[51] as well as the preservation of denominational identity and integrity were enforced throughout the event thanks to the conference structure:[52] in the morning, each Christian group conducted its own regular conference sessions in separate small conferences across the city, while, in the afternoon, all the sub-conferences contributed workshops for all to attend until the evenings, when plenary sessions in the stadium gathered all the participants. The importance of the manifestation of the gifts of the Spirit was highlighted through the idea of a "word gifts unit", meaning that all the planning committee members had access to the microphones along with a few trusted leaders from sub-conferences to prophesy. Ralph Martin's prophecy *The Body of My Son is broken*[53] and his focus on the sinfulness of the disunity of the church and the need for repentance and intercession to make amendment represented the peak of all the Spirit revelations,[54] while the "unforgettable scenes" of Cardinal Suenens sharing the platform with Thomas Zimmerman, head of the Assemblies of God, and James O. Patterson, head of the Church of God in Christ, were the apex of the ecumenical success of the conference and symbolized the overcoming of historical denominational barriers.

About 50,000 Christians attended the Kansas City conference. However, the proportion of preregistered participants immediately shows not only how the Catholics and non-denominational charismatics took center stage, but also how deep the division was among non-denominationals as a result of the Shepherding movement's controversy:[55] 48.73% Catholic; 17.56% non-denominational B; 10.23% non-denominational A; 5.93% Lutheran; 5.69% Episcopalian; 3.28% Presbyterian; 2.51% unassigned; 1.79% United Methodist; 1.59% Mennonite; 0.90% Denominational Pentecostals; 0.52% Messianic Jewish. As a matter of

fact, originally, there was to be only one non-denominational group, co-led by Mumford and Frost, but eventually, the ongoing tensions over the discipleship question resulted in another group whose leaders, such as Robertson or Ralph Wilkerson (Melodyland Christian Center) or Ralph Mahoney (World MAP), did not want to share the platform with CGM leaders.[56] Non-denominational group B was finally led by Fort Lauderdale leaders and attracted the second largest delegation to the conference, with over 12,000 persons. Interestingly, during the conference preparation period, it was pretty clear how weights and measures would have eventually been distributed, since a note by the Ecumenical Council details: "A. Rumors: some non-Catholics claim Conference is a Vatican plot. B. Some Catholics and others fear a Ft. Lauderdale plot".[57]

Despite everything, the conference was a success in terms of numbers, visibility, and cooperation, and it had significant ramifications.[58] For example, further ecumenical Jesus rallies (i.e., in Washington in 1980) and conferences (i.e., New Orleans in 1986) took place throughout the United States in the years that followed; denominations such as the United Methodists and Baptists institutionalized their own charismatic organizations for the first time after Kansas City; three new service committees—Pentecostal, United Church of Christ and Holiness/Wesleyan/Nazarene—were established to provide services for charismatics.[59] Positive conclusions on Kansas City were also elicited from leaders of the Council:

"III. Conclusions from Kansas City: A. Derek listed four things in his newsletter.1. Kansas City proved that leaders from varying backgrounds can cooperate; 2. Cooperation produces far greater impact than individual ministry; 3. We are interdependent—we need each other; 4. Ecumenical gatherings bless all who participate. B. The General Council concludes: 1. God has given us a mandate to become one; 2. Kansas City was a forerunner of a greater unity; 3. The unity of the Kansas City Conference grew out of our unity; 4. Unity, then, can be imparted—even as it was from our pilgrimage".[60]

However, the motto "at Kansas City the Lord called us to reach beyond our denominational walls to work and pray aggressively for a higher goal—the unification of all Christianity"[61] was not well received within Roman Catholicism. Such an event, where Christians from different churches and affiliations came together to celebrate a charismatic spirituality, created ecumenical expectations within some Catholic charismatic groups that eventually preoccupied ecclesiastical hierarchies, particularly the Vatican, which, in contrast to the charismatic participants, perceived some of the stances expressed at the Kansas City Conference as more non-denominational than ecumenical.[62] Pan-charismatic ecumenism contributed to the rise of an alarmist attitude toward the CCR among ecclesial authorities in the United States and in Rome during the delicate phase of legitimization of the movement within the Catholic Church[63] to the point of an alleged threat of a condemnation of the CCR in 1978, when the Congregation for the Doctrine of the Faith (CDF) seemed to have prepared an opposition document and asked for clarification, especially with regard to ecumenical forms lived within certain covenant communities.[64] Thanks to Suenens' timely intervention, the situation was resolved, but only for a while. With John Paul II's pontifical election (1978) and Suenens being replaced by Paul Josef Cordes as episcopal adviser of the CCR (1984), a new and different season for Catholic charismatics would start, that of "Catholicization" at the expense, among other things, of ecumenical experimentations and networking.[65]

## 5. Conclusions

In the late 1960s and 1970s, ecumenical relationships—some would call them contaminations—between charismatic groups in the United States were ordinary. Although Catholic charismatic apologetic literature at the beginning diminished the role of such "cross-pollination" in order to support the ongoing legitimization process of the movement within the Catholic Church, ecumenical relationships undoubtedly shaped the Catholic renewal as a whole.

Pulkingham's pastoral influence, the Ecumenical Council between Catholics and non-denominational charismatics, and the Kansas City conference were important elements

within this charismatic ecumenism—or inter-denominationalism/non-denominationalism, if the Vatican perspective is adopted. They demonstrate how Catholic charismatics stepped beyond the borders of their church to experience and learn about charismatic spirituality, in order to eventually translate it into Catholic language and practice. This did not always please ecclesiastical hierarchies and the Vatican, as exemplified in the influence of the Shepherding movement within Catholic charismatic communities, at least those which emerged from the Ann Arbor/South Bend model, which, in some measure, contributed to a focus on authority, emphasizing vertical community leadership/eldership, submission and servanthood;[66] or the pan-charismatic attitude of the Kansas City conference that was perceived too loose and dangerous for Catholic integrity.

Ecumenical grassroots contacts among churches and denominations in the late 1960s and 1970s, as explored here, acquire an important meaning for the history of the CCR in subsequent years, not only because ecumenism was an object of attention for all the actors involved, both in the United States and in Rome,[67] but also because such ecumenical relationships would become the litmus test for determining the Catholic orthodoxy of the movement. In the 1980s, in fact, some of these experiences that derived from ecumenical contacts would have to be abandoned for the sake of a full integration within John Paul II's Catholic Church.

**Funding:** This article is an offshoot of the project 'The Catholic Charismatic Renewal (CCR): An Historical Analysis Between US and Europe' (CAT-CAM), which has received funding from the European Union's Horizon 2020 research and innovation programme under the Marie Skłodowska-Curie grant agreement no. 654994.

**Institutional Review Board Statement:** The study was conducted according to the guidelines of the Declaration of Helsinki, and approved by the Ethics Committee of the European Commission under the grant agreement no. 654994.

**Informed Consent Statement:** Not Applicable.

**Data Availability Statement:** Not Applicable.

**Conflicts of Interest:** The author declares no conflict of interest.

## Notes

1　Cfr. Valentina Ciciliot. "The Origins of the Catholic Charismatic Renewal (CCR) in the United States: Early Developments in Indiana and Michigan and the Reactions of the Ecclesiastical Authorities". *Studies in World Christianity*. 25/3 2019, pp. 250–73 (Ciciliot 2019) and Valentina Ciciliot. "From the United States to the World, Passing through Rome: Reflections on the Catholic Charismatic Movement". *PentecoStudies*. 19/2 2020, pp. 127–15 (Ciciliot 2020).

2　The most recent works on the CCR are Andrew Atherstone, Mark Hutchinson and John Maiden eds. *Charismatic Renewal in Europe and the United States Since 1950*. Leiden: Brill, forthcoming (Atherstone et al. forthcoming); Alan Schreck. *A Mighty Current of Grace: The Story of the Catholic Charismatic Renewal*. Frederick: The Word Among Us Press, 2017 (Schreck 2017), and Susan A. Maurer, *The Spirit of Enthusiasm. A History of the Catholic Charismatic Renewal, 1967–2000*. Lanham: University Press of America, 2010 (Maurer 2010). Among memoirs and first studies: Kevin Ranaghan and Dorothy Ranaghan. *Catholic Pentecostals*. Paramus: Paulist Press, 1969 (Ranaghan and Ranaghan 1969); Edward O'Connor. *The Pentecostal Movement in the Catholic Church*. Notre Dame: Ave Maria Press, 1971 (O'Connor 1971); Joseph H. Fichter. *The Catholic Cult of the Paraclete*, New York: Sheed and Ward, 1975 (Fichter 1975); René Laurentin. *Catholic Pentecostalism*. Garden City: Doubleday & Company, 1977 (Laurentin 1977); Meredith B. McGuire. *Pentecostal Catholics. Power, Charisma and Order in a Religious Movement*. Philadelphia: Temple University Press, 1982 (McGuire 1982); Richard J. Bord and Joseph E. Faulkner. *The Catholic Charismatics. The Anatomy of a Modern Religious Movement*. University Park: Pennsylvania University Press, 1983 (Bord and Faulkner 1983); Patty Gallagher Mansfield. *As by a New Pentecost. The Dramatic Beginning of the Catholic Charismatic Renewal*. Steubenville: Franciscan University Press, 1992 (Gallagher Mansfield 1992).

3　Ciciliot. "The Origin of the Catholic Charismatic Renewal", pp. 252–57.

4　The most famous example is "The Alternative Jesus: Psychedelic Christ." *Time Magazine*. Monday, 21 June 1971 ("The Jesus Revolution", cover story). See also the earlier Mary Papa. "Dropping in on Notre Dame's Pentecostals. People Having a Good Time Praying." *National Catholic Reporter*. 17 May 1967. Edward O'Connor quoted most of the magazines and newspapers that covered the events at Notre Dame in "The literature of the Catholic Charismatic

Renewal 1967–1974." In *Perspectives on Charismatic Renewal*. Edward O'Connor ed., Notre Dame/London: University of Notre Dame Press, 1975, pp. 145–15 (O'Connor 1975a). Noteworthily, a direct testimony reported that "in spring and summer 1967 at ND we prayed over hundreds and hundreds of students and a wide variety of people from all over the country. This wasn't our choice, but it happened because of the national publicity. [ . . . ] The CCR could have continued to expand gradually one-to-one, but the ND experience generated a huge amount of national publicity and got thousands of people interested within a few months. Before ND, it was more like, a friend telling a friend. After ND, it was more like, shout it from the rooftops and make headlines. We didn't plan that; it just happened because we were at ND. Prayer meetings at Duquesne didn't make national news. Prayer meetings under the ND golden dome did." Tom Noe, written communication. 5 May 2018.

5　Doug Wead. *Catholic Charismatics. Are They for Real?* Carol Stream: Creation House, 1973, p. 5 and also pp. 1–14, pp. 105–107, and appendix, pp. 108–120 (Wead 1973). He also wrote *Father McCarthy Smokes a Pipe and Speaks in Tongues*. Norfolk: Wisdom House, 1972 (Wead 1972). O'Connor mentions Wead's participation in *The Pentecostal Movement*, pp. 64–65. Doug Wead's biography at: http://www.dougwead.com/biography.html (accessed on 16 May 2021). Not by chance, "Catholic Pentecostals" was later replaced by the term "Catholic charismatics".

6　Wead. *Catholic Charismatics*, p. 10.

7　More details on the historical events in the Midwest in Valentina Ciciliot. "The Origins of the Catholic Charismatic Renewal in the United States: The Experience at the University of Notre Dame, South Bend, Indiana." In *Charismatic Renewal in Europe and the United States Since 1950*, pp. 144–64. Forthcoming. (Atherstone et al. forthcoming).

8　Interestingly, the role of the Cursillo movement of the 1950s and 1960s in the United States and its specific contribution to the rise of Catholic charismatic prayer groups was not highlighted by the early Catholic historiography of the CCR. On the link between the Cursillo and Catholic charismatic leaders see Ciciliot. "The Origins of the Catholic Charismatic Renewal in the United States," pp. 148–52.

9　Elena Guerra (1835–1914) was the founder of the Oblates of the Holy Spirit. Because of her emphasis on devotion to the Holy Spirit and her commitment to spread it, she was considered the one who prepared the ground within the church for the Catholic charismatic movement. See Peter Hocken. *The Catholic Charismatic Renewal, in The Century of the Holy Spirit. 100 Years of Pentecostal and Charismatic Renewal 1901–2001*, Vynson Synan ed. Nashville: Thomas Nelson Publishers, 2001, pp. 212–13 (Hocken 2001).

10　One of the best examples is theologian Edward O'Connor's work. See, in particular "The Hidden Roots of the Catholic Charismatic Renewal in the Catholic Church." In *Aspects of Pentecostal-Charismatic Origins*, Vinson Synan ed. Plainfield: Bridge-Logos 1975, pp. 168–91 (O'Connor 1975b), where he does not stress ecumenical interconnections between charismatics and he enlists the ecumenical movement along with other movements that he considers important for the emergence of the Catholic charismatic renewal such as the biblical, the lay, the liturgical and also the mystical body. Thus, it is legitimate to ask how much the frequent appeal to Vatican II in the first charismatic theological and historical literature was instrumental in the legitimization of the CCR and how much the council itself actually contributed to the birth of the renewal.

11　Ciciliot. *From the United States to the World, Passing through Rome*, pp. 139–44.

12　Classic is the book of Episcopalian Dennis J. Bennett. *Nine O'Clock in the Morning*. Plainfield: Logos International, 1970 (Bennett 1970). Cfr. Also *The Century of the Holy Spirit.*

13　See William Storey's manuscript of the Duquesne weekend retreat written in 1994, Duquesne University Archives and Special Collections, Catholic Charismatic Movement, Subject Files.

14　James Manney. "Before Duquesne: Sources of the Renewal." *New Covenant*. 1973 February 12–17. See also Jim Cavnar and Peter Collins. "The Pre-history of the Duquesne Weekend." 2 September 2014 (paper sent from Jim Cavnar via email, 12 June 2018).

15　Michael Harper. *Three Sisters*. Wheaton: Tyndale House, 1979, p. 29 (Harper 1979).

16　The Ranaghans describe the meeting at FGBMFI chapter president Ray Bullard's house with these words: "Here we are, a group of Roman Catholics, formed in the spiritual and liturgical traditions of our Church, all university trained 'intellectual types'. The people with whom we were meeting were mostly from an evangelical background. They spoke with a scriptural and theological fundamentalism that was foreign to us. Furthermore, the way they spoke and prayed, the type of hymns they sang—all was so different that at first it was very disturbing. On the natural level these 'cultural' differences were more than enough to keep us far apart from each other. Yet, in spite of these personal differences, we were enabled to come together in common faith in Jesus, in the one experience of his Holy Spirit, to worship our Father together", Ranaghan. *Catholic Pentecostals*, p. 41. See also *First Notre Dame-F.G.B.M.F meeting and results, 13 March 1967*, in University of Notre Dame Archives (UNDA), Edward O'Connor Papers (EOC), 1/01, Charismatic Renewal 1967 (it is the tape transcription of the three-hour meeting). «Voice» dedicated space to the

Catholic charismatics. See Edward O'Connor, "Pentecost at Notre Dame." *Voice*. July–August 1967, pp. 25–29 or "Gentle Revolution. The Catholic Pentecostal Movement in Retrospect." *Voice*. September 1971, pp. 3–7.

17      In fact, a 'Catholic-centric' historiography has neglected to underline the ecumenical networks between charismatics. See as examples O'Connor. *The Pentecostal Movement in the Catholic Church* and Schreck. *A Mighty Current of Grace*.

18      True House formally ended in 1975, with some of its members joining the People of Praise or other communities, whereas the People of Praise and the Word of God are still in existence.

19      Julia Duin, *Days of Fire and Glory. The Rise and the Fall of a Charismatic Community*. Baltimore: Crossland Press, 2009 (Duin 2009). Pulkingham's influence on the Jesus People in Larry Eskridge. *God's Forever Family. The Jesus People Movement in America*. Oxford: Oxford University Press, 2013, p. 122 (Eskridge 2013).

20      Michael Harper. *A New Way of Living: How the Church of the Redeemer, Houston, Found a New Life-Style*. Plainfield: Logos International, 1973 (Harper 1973).

21      Duin. *Days of Fire and Glory*, 97. Pulkingham also worked together with Catholic charismatics in other editorial activities such as in the essay collection Ralph Martin ed. *The Spirit and the Church: A Personal and Documentary Record of the Charismatic Renewal and the Ways It Is Bursting to Life in the Catholic Church*. New York: Paulist Press, 1976 (Martin 1976), among Cardinal Léon-Joseph Suenens, theologian George T. Montague and president of the University of Steubenville Michael Scanlan.

22      Ibid. pp. 74–75. Cardinal Léon-Joseph Suenens visited Church of the Redeemer in February 1974. Ivi, pp. 134–35. See *Rétrospective de mon voyage aux Etats-Unis*, in Archdiocesan Archives Mechelen (AAM), Archivum L. J. Suenens, box 26, Voyage aux Etats-Unis, 26 janvier-11 février 1974, Correspondance, Textes, Presse. Suenens wrote: "Je crois que Houston es tune image, un prototype de ce qui serait un christianisme vécu dans l'authenticité", p. 3.

23      See UNDA, EOC, 1/04, Charismatic Renewal 1969, *Grace and Peace*, August 19, 1969 (it is a sort of report from True House community).

24      Cfr. Tom Noe. "Notes from Early Community Meetings. September, 1971." *People of Praise Vine & Branches*. September 1996, p. 5 and "Notes from Early Community Meetings. Making the Covenant." *People of Praise Vine & Branches*. October 1996, pp. 6–7.

25      Eventually, it seems that Pulkingham had a "blow-out" with Clark and Martin over women in leadership, resulting in no more invitations to the Notre Dame conference. Ivi, p. 139. The preference for male leadership and restricted traditional roles for women are well explained in Clark's further book *Man and Women in Christ: An Examination of the Roles of Men and Women in Light of Scripture and the Social Sciences*. Ann Arbor: Servant Books, 1980 (Clark 1980), as Pulkingham's radical idea of "brother in Christ" and his egalitarian family theology was widely well-known at that time.

26      Their biographies in David S. Moore. *The Shepherding Movement: Controversy on Charismatic Ecclesiology*. London-New York: T&T Clark International, 2003, chapter 3, pp. 33–45 (Moore 2003).

27      David S. Moore. "Shepherding Movement." In *The New International Dictionary of Pentecostal and Charismatic Movements*. Stanley M. Burgess and Eduard M. van der Maas eds. Grand Rapids: Zondervan, 2002, pp. 1060–62 (Moore 2002). See also Harper. *Three Sisters*. 91-96. See also the journalistic book Sara Diamon. *Spiritual Warfare: The Politics of the Christian Right*. Montréal/New York: Black Rose Books, (Diamon 1990, chapter 4, pp. 111–46).

28      Moore. *The Shepherding Movement*, p. 2.

29      Cfr. Don Basham. "Forum: CGM and New Wine." *New Wine*. December 1976, p. 319.

30      Harper. *Three Sisters*, p. 30.

31      i.e., "Charismatic Movement is Facing Internal Discord over a Teaching Called 'Discipleship'." *New York Times*. 16 September 1975, and Edward E. Plowman. "The Deepening Rift in the Charismatic Movement." *Christianity Today*. 10 October, 1975, pp. 52–54.

32      Moore. *The Shepherding Movement*, pp. 121–22.

33      Ibid. 123.

34      Moore. *The Shepherding Movement*, 121. Cfr. Also Kilian McDonnell ed. *Seven Documents on the Discipleship Question*. In *Presence, Power, Praise: Documents on the Charismatic Renewal*. Vol. 2. Collegeville: Liturgical Press, 1980, pp. 116–47, particularly 122 (McDonnell 1980).

35      Moore. *The Shepherding Movement*, pp. 115–16.

36      "Logos Report: National Men's Shepherds Conference." *Logos*. November 1975.

37      Pictures of its members in Moore 126. The Ecumenical Council minutes are out of consultation, although some of them are also available at the Bentley Historical Library, University of Michigan, Tom Yoder Papers, 1967–1991. Diamond wrote that the Council's "purpose was to strengthen the shepherding system across the denominational lines. By the mid-1970's, the Council had expanded to include Catholic shepherding stalwarts, Paul DeCelles and

Kevin Ranaghan and also Larry Christianson, a leader in the Lutheran charismatic renewal", Diamond. *Spiritual Warfare*, pp. 122–23.

38    Minutes of the Meeting of the Council, 17–19 December 1975, Ann Arbor, "Relations between communities. We will move away from the parallel development of two networks of communities toward one network of communities. We will begin to have closer fellowship among our communities. We encourage visits between our communities. We will all study the statement of community order. [ . . . ] The Council delegates Derek and Steve to work in East Lansing with the two communities to bring about some kind of organic unity (one body with one government)".

39    Minutes of the Meeting of the Council, 8–10 September 1974, Ft. Lauderdale, "Speaking about the Council. We will not make a public announcement about our commitment together".

40    As an example, "The following will be the relationships of personal subordination within the council: Derek, John and Charles to Bob; Bob, Ern and Don to Derek; Ralph and Steve as previously", in Minutes of the Meeting of the Council, 8–10 September 1974, Ft. Lauderdale.

41    Minutes of the General Council Meeting, 8–10 August 1977, Ann Arbor. See also Minutes from the Meetings of the General Council Held During Pilgrimage, May/June, 1977.

42    These tensions are partially explained in Ciciliot. *From the United States to the World*, n. 32, p. 140.

43    For the later history of the Shepherding movement, after its zenith in 1977, see Moore. *The Shepherding Movement*, chapter 9, pp. 154–78.

44    Minutes of the meeting of the Ecumenical Council, February 5th, 6th, and 7th, 1985, in Fredericksburg, Virginia.

45    Ciciliot. *From the United States to the World*, p. 139.

46    It was never widely known that the idea of the Kansas City conference and its implementation originated from the Ecumenical Council, although evidence shows this had been promoted there since 1974. See Minutes of the Meeting of the Council, 15–17 December 1974, Ft. Lauderdale: "We will consider a 'Three Rivers' general conference in 1977. Kevin Ranaghan will present a proposal to us". In fact, according to one of Synan's symposium reports, the idea of the conference was born around 1974 when Synan, Russell Spittler and Rodman Williams discussed the fact that denominational charismatic organizations held separated annual meetings every year, Vinson Synan. "The 1977 Kansas City Conference: A Study in Ecumenical Sensitivity." In *Ecumenism and Charismatic Renewal.* Diocesan Liaison Theological Symposium, Sacred Heart Retreat House, Sedalia, Colorado, 26–28 September 1986, p. 2. Moore explained that "perhaps the leaders wanted to avoid bringing unnecessary controversy to the conference since the five Shepherding leaders were so much part of the Ecumenical Council. Whatever the reason, the group seldom commented publicly on their roles in the Kansas City conference", Moore. *The Shepherding Movement*, p. 132. It is also possible that different charismatic leaders were thinking about such an event. Interestingly, Clark explained in 1978, in a letter that Moore holds privately, that "at that time, the Catholic Charismatic Renewal Service Committee [a.n., of which Clark and Ranaghan were members] objected to the plans because they did not rest upon a leadership group that was properly constituted and would understand how to work in an ecumenical way" and he also stressed the fact that the conference was ecumenical and not nondenominational. Two insights emerge here. First, it was difficult for the CCRSC to understand the ecumenically broad scope of Kansas City as it was being theorized by the two Ann Arbor leaders with their organizational convergence with the Fort Lauderdale leaders—a convergence, however, that was not merely instrumental, but was deep and based on sharing of spiritual and pastoral authority and charismatic leadership training. Second, Martin and Clark's belief that they were acting in the fullest sense of catholicity, but that was not the opinion of the Vatican, which soon perceived Kansas City as a conference more 'interdenominational' than ecumenical and pushing forward a distinctly different form of Catholic ecumenism. See also Ciciliot. *From the United States to the World*, 136, pp. 140–41.

47    Ralph Martin. "The Mighty Stream of God." *New Wine*. November 1974, pp. 14–17.

48    See Vinson Synan. "Kansas City Conference". Burgess and Van Der Maas eds. *The New International Dictionary of Pentecostal Charismatic Movements*, p. 816. For media coverage see John Blattern. "A Living Prophecy: Report on the Conference." *New Covenant*. September 1977, pp. 4–9; "Charismatic Unity in Kansas City." *Christianity Today*. 12 August 1977, pp. 36–37; "Kansas City Conference Demonstrates Unity." *ICO Newsletter*. November 1977, first three pages; "Charismatic Renewal: Up to Date in Kansas City." *America*. September 24, 1977, pp. 164–66; Jason Petosa. "Suenens calls gathering ecumenical triumph." *National Catholic Reporter*. August 12, 1977, pp. 1, 4.

49    Synan. "Kansas City Conference."816, and Id. "The 1977 Kansas City Conference", p. 3. The list of names is shorter in *The New International Dictionary of Pentecostal Charismatic Movements*.

50    Synan. "The 1977 Kansas City Conference", p. 5.

51    Synan used the term "cross-polinization" [sic!], in Ibid. p. 7.

52    As an example of the concern for integrity, separated masses for Catholic participants were scheduled.

53    Synan. "The 1977 Kansas City Conference", p. 14.

54    Ralph Martin. "The Sin of Disunity. How Can We Heal the Broken Body of Christ?" *New Covenant*. November 1977, pp. 16–18.

55    Synan. "The 1977 Kansas City Conference", 10. These were also the actual proportions of conference attendants, with minor changes. For similar statistical data see also General Council Meeting, 8–10 August 1977, Ann Arbor, MI.

56    Moore. *The Shepherding Movement*, pp. 133–34.

57    The General Council Held During Pilgrimage, May/June, 1977.

58    As an example of political visibility, Ruth Carter Stapleton, U.S. President Carter's sister, was one of the major speakers in the Baptist sub-conference, and the president himself sent a telegram asking all the participants to pray for him, "A Charismatic Time Was Had by All." *Time Magazine*. August 8, 1977. See also Synan. "The 1977 Kansas City Conference", p. 15.

59    John Blattern. "A Living Prophecy." *New Covenant*. September 1977, p. 5.

60    Minutes of the General Council Meeting, 8–10 August 1977, Ann Arbor.

61    "From the Editor" (Bert Ghezzi). *New Covenant*. October 1977.

62    It seems that Jean Jérôme Hamer (secretary of the CDF from 1973 to 1984) used the term "interdenominational" to describe Ann Arbor- like covenant communities after Kansas City. See CUAA, NCCB, box 120, Ad Hoc Committee: Catholic Charismatic Renewal, 1978-1979, Letter from Thomas Kelly, associate general secretary of the NCCB from 1972 to 1977 and of the USCCB from 1977 to 1982, to Charron, July 12, 1978: Kelly reported his trip to Rome and his conversation with Hamer, who was concerned about "the very tight organization of some of the Charismatics e.g. Ann Arbor, it would prevent the Bishop from exercising his responsibility as magister vitae spiritualis. [...] Finally, he feels that the new communities are quite improperly called 'ecumenical', he sees them rather as 'interdenominational'".

63    Concerns about ecumenism and other practices of Catholic charismatics were evident well before the Kansas City conference. See Bentley Historical Library (BHL), University of Michigan, Tom Yoder Papers, 1967-1991, box 5, WOG (Word of God) Archives, Documents 1974–1975, Letter from McDonnell to Clark, 4 November 1974 (on Hamer's conversation with McDonnell) and Memo to Service Committee from McDonnell, on Ecumenical Dimensions of the Renewal, 3 December 1974: "There is no doubt that the No. 1 cause of anxiety is the ecumenical aspects of the renewal. If Rome or national hierarchies issue statements which contain grave reservations with regard to the renewal, it will very likely be because of these ecumenical dimensions".

64    CUAA, NCCB, box 120, Ad Hoc Committee: Catholic Charismatic Renewal, 1978-1979, Ad Hoc Committee on the Catholic Charismatic Renewal, Report of Meeting, March 9, 1978; Letter from Suenens to Frey, February 9, 1978; Letter from Quinn to Your Eminence (Cardinal Franjo Seper, CDF), March 17, 1978; Letter from Seper (?) to Quinn, April 19, 1978. Cfr. also AAM, Archivum L.-J. Suenens, Renouveau charismatique, Correspondance Suenens, Evénement importants, box 85, folder Correspondance importante, Correspondence between Jean-Marie Villot (secretary of State), Suenens, and Paul VI, avril 1978; Renouveau charismatique, Ann Arbor, Word of God, Ralph Martin, Kevin Ranaghan, Steve Clark, Kilian McDonnell, box 87, folder Ralph Martin (1), Notes of the meeting at the Holy Office on 18 October 1977. Interesting information also in the interviews with Ralph Martin 10/07/2018 and Steve Clark 12/07/2018 (not recorded).

65    Ciciliot. *From the United States to the World*, particularly pp. 143–44.

66    Convergences existed if you think about the concept of headship and submission theorized by Steve Clark in *Building Christian Communities: Strategy for Renewing the Church*. Notre Dame: Ave Maria Press, 1972 (Clark 1972). See also Steve Clark. *Unordained Elders and Renewal Communities*. New York: Paulist Press, 1976 (Clark 1976). However, it is necessary to always take into account the traditional Catholic reliance on authority as an important historical element.

67    The second Malines document is not by chance on ecumenism, Léon-Joseph Suenens. *Ecumenism and Charismatic Renewal. Theological and Pastoral Orientations*. Malines Document 2. Ann Arbor: Servant Books, 1978 (Suenens 1978). Cordes also insisted on this topic in his *Charismatic Renewal: A Balancing Force in the Church Today*. South Bend: Greenlawn Press, 1985 (Cordes 1985) and *Call to Holiness. Reflections on the Catholic Charismatic Renewal* Collegeville: Liturgical Press, 1997 (Cordes 1997).

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
