# Peer review of "“Pray Aggressively for a Higher Goal—The Unification of All Christianity”: U.S. Catholic Charismatics and Their Ecumenical Relationships in the Late 1960s and 1970s"

_religions, doi:10.3390/rel12050353_

Round 1

Reviewer 1 Report

This is a very helpful and well written piece of scholarship which describes the complex ecumenical relationships and dynamics within the charismatic movement in the United States. It draws upon an exemplary range of primary sources and in the main relates the argument to the existing scholarship effectively.

The following might be considered:

P. 2 More might be said here about the contribution of Cursillo alongside the adoption of pentecostal/charismatic Protestantism. 

P. 4 asserts that The Word of God was influenced by the Jesus movement. This point might be elaborated on, and some supporting evidence/examples offered.

p. 4-5 It is perhaps worth saying more about the 'High'/sacramental emphasis of Church of the Redeemer. Did this lend itself to some affinity with Roman Catholics?

p. 5 ' provides a realistic glimpse’ – slightly unclear. Revise.

p. 9 Was CRS still strictly speaking the CCRSC corporation by this point? Or was there by then the trustees divided between TWOG, PoP and x2 members of CCRSC. Minor difference in practice, perhaps - but worth checking and clarifying if necessary.

Author Response

Comments on file.

Reviewer 2 Report

Line 18-27 & elsewhere: Problematic language at the beginning of the essay, which traces the origins of the movement, suggests some spiritual omniscience on the part of the author. Whether or not they were actually baptized by the Holy Spirit is impossible to prove, even by the participants themselves. In its place, author should use more objective language: Kiefer, Storey et. al. claimed to be baptized by the Holy Spirit… 

Line 49 — elaborate on this historic animosity — was it social or merely theological?

Lines 52-67 — a lot of missing context here. Who was Elena Guerra (a general reader, lamentably, has little idea why she would be a precursor to the CCR); Speaking of origin myths, the article seems to ignore the role of the Cursillo Movements of the 1950s and 1960s. To be sure, the argument here is that movements such as this should be seen as part of the broader “Catholicization" of small-p pentecostalism. But, the elements of that origin myth need to be clear if the goal of the essay is to, in part, explain the myth's limitations. 

The argument of the essay, compelling as it is, only becomes clear after page 3 and needs significant strengthening. Lines 99-101 need a footnote that offers a thorough accounting of the more "Catholic-centric" historiography." Until that point, it only promises to "highlight the experiences" of ecumenical encounters. 

Line 49: Please elaborate on "historic animosity." Several points in the article need better explanation of context and definition of terms for a general reader. See "Jesus Movement," for example. Line 125

Lines 89-90: An additional point of elaboration needed regarding the narrative and demographic  profile of charismatics, particularly the Businessmen’s Fellowship. What does this say about ideological tensions in the church?  How did charismatics merge faith and “business practices”? How might this challenge or accommodate a Catholic critique of free market ideology, prevalent in the 1950s? 

Line 106-107 Say more: what was this crisis and long-term consequences for both the CCR and the larger Catholic ecumenical spirit after Vatican II? How important was the CCR, and Vatican responses to it, for setting the limits of ecumenical encounter? How important was it to shaping the intersections of Catholic and Protestant socio-political alliances in the 1970s and beyond? Only passing menion (in a quote; lines 41-42) is made to political sensibilities, but attention to demographics seems necessary. This is especially relevant as a member of POP has recently risen to the Supreme Court. 

Line 125; Another reference that needs definition. If they were at all influenced by the Jesus movement and/or the counterculture, the audience needs more explanation for the sake of a complete story. This was a significant, if broad, influence that the author assumes more they demonstrate. 

138 — How? The author leaves a lot of unfinished stories that a general audience remains lost without. It also drops many names without thorough discussion of their relationships. Try focusing on the significant players for the purposes of this narrative and creating clear connections among them. 

150-155 — cut down on extensive quotes from secondary sources; Suggests that the author takes a lot of the existing, and less-self critical insider, literature of the movement at face value. 

Line 165: What makes it “realistic”? To whom? Focus on what it reveals about the community and its own origin story. Who was the author and what perspective or goals for the movement does it reveal? Much more contextual analysis needed here.

Lines 220-230: this is obviously an important narrative of the internal controversies, but the organization of the essay (along the lines of different movements) obscures what was at stake here. How did the Shepherding movement represent controversies and competing meanings of "authority" and "discipleship" as a whole throughout the CCR's generative period? How did it present across these pages of its various media? Offer a fuller reading of these sources. In this way, the manuscript might distinguish itself from these other sources in the historiography. 

Line 220-222 Clear to whom? Avoid passive voice throughout, as it obscures the actors and stakeholders in these debates. 

Line 233-265: This massive paragraph does a lot of heavy lifting for the article as a whole, but it covers a number of different topics. Is the weight of the argument simply that Catholics were "present" at these meetings and event hosted them? What about its deeper significance? How did these Catholics understand their role here, vis-a-vis both the larger CR and the Catholic hierarchy? 

Line 265-282: Too much of the primary source and not enough interpretation. It's still unclear what this shows. 

Line 290-293: What was at the center of this tension? This seems significant!

A closer reading of the KC conference's messages needed here. 

Author Response

Comments on file.

Round 2

Reviewer 2 Report

I agree with and understand your response. I also appreciate the willingness to clarify certain points in my original comments. Thanks for offering context on future work here.